# Experimental Analysis of the Use of Cranial Electromyography in Athletes and Clinical Implications

**DOI:** 10.3390/ijerph19137975

**Published:** 2022-06-29

**Authors:** Alessio Danilo Inchingolo, Carmela Pezzolla, Assunta Patano, Sabino Ceci, Anna Maria Ciocia, Grazia Marinelli, Giuseppina Malcangi, Valentina Montenegro, Filippo Cardarelli, Fabio Piras, Irene Ferrara, Biagio Rapone, Ioana Roxana Bordea, Dario Di Stasio, Antonio Scarano, Felice Lorusso, Andrea Palermo, Kenan Ferati, Angelo Michele Inchingolo, Francesco Inchingolo, Daniela Di Venere, Gianna Dipalma

**Affiliations:** 1Department of Interdisciplinary Medicine, University of Bari “Aldo Moro”, 70124 Bari, Italy; ad.inchingolo@libero.it (A.D.I.); pezzollacarmen@gmail.com (C.P.); assuntapatano@gmail.com (A.P.); s.ceci@studenti.uniba.it (S.C.); anna.ciocia1@gmail.com (A.M.C.); graziamarinelli@live.it (G.M.); giuseppinamalcangi@libero.it (G.M.); valentinamontenegro@libero.it (V.M.); drfilippocardarelli@libero.it (F.C.); dott.fabio.piras@gmail.com (F.P.); ire.ferra3@gmail.com (I.F.); angeloinchingolo@gmail.com (A.M.I.); daniela.divenere@uniba.it (D.D.V.); giannadipalma@tiscali.it (G.D.); 2Department of Oral Rehabilitation, Faculty of Dentistry, Iuliu Hațieganu University of Medicine and Pharmacy, 400012 Cluj-Napoca, Romania; 3Multidisciplinary Department of Medical-Surgical and Dental Specialties, University of Campania Luigi Vanvitelli, Via L. De Crecchio 6, 80138 Naples, Italy; dario.distasio@unicampania.it; 4Department of Innovative Technologies in Medicine and Dentistry, University of Chieti-Pescara, 66100 Chieti, Italy; ascarano@unich.it (A.S.); felice.lorusso@unich.it (F.L.); 5Implant Dentistry College of Medicine and Dentistry Birmingham, University of Birmingham, Birmingham B4 6BN, UK; andrea.palermo2004@libero.it; 6Faculty of Medicine, University of Tetovo, 1220 Tetovo, North Macedonia; kenan.ferati@unite.edu.mk

**Keywords:** balance, dental occlusion, sports, splints, occlusal splints, posture, electromyography

## Abstract

Background: Cranial surface electromyography is assumed to analyze the correlation between the stomatognathic apparatus and the muscular system and its implications on the physical status of professional athletes. The purpose of this paper is to evaluate surface cranial electromyography as an aid in the diagnosis and treatment of occlusal and muscular disorders in professional athletes. Methods: A sample of 25 athletes (mean age 23 years, 20 men and 5 women) underwent electromyographic recording; among them, 13 had a sports injury condition (symptomatic athletes), while 12 were in perfect physical condition (asymptomatic athletes). At odontostomatological examination, 6 showed cranio-mandibular disorders (dysfunctional athletes), while 19 showed no disorders (functional athletes). The treatment plan to resolve the symptoms of the dysfunctional athletes was chosen based on the electromyographic data. One month after the start of therapy with an occlusal splint, a follow-up was performed, and the results were compared with the initial data. Results: Statistical analysis showed that the chosen therapy following the use of electromyography was effective in 72% of cases, while 28% of patients did not respond to therapy (*p* = 0.028). Conclusions: The use of cranial electromyography in competitive athletes is a valuable tool in therapeutic choice aimed at balancing occlusal loads and improving the patient’s global tonic postural attitude, resulting in positive feedback in the qualitative assessment of sports performance.

## 1. Introduction

Cranial electromyographic analysis examines the correlation between the stomatognathic apparatus and the muscular tonic system. The stomatognathic apparatus comprises an anatomical district consisting of all those structures capable of supporting a series of biological functions (e.g., phonation, mastication, swallowing, etc.) that are carried out through a closed kinematic chain, realized by neck muscles, tongue, and masticatory muscles, fixed to a skeletal component formed by the skull, jaw, and cervical spine [1,2,3,4,5].

Discordant data are reported in the literature regarding whether or not there is a correlation between occlusion and posture [6].

Recently, it has been proposed that the influence of dental occlusion on postural control depends on external disturbances. In particular, dental occlusion contributes more to postural control in the presence of more difficult conditions, such as unstable conditions and muscle fatigue [7,8,9].

However, it was recently suggested that the stomatognathic system may potentially play a role in postural balance in 2% of people [10]. This contribution may appear to be negligible in terms of the overall population’s physical ability, but it could play a significant role in athletic performance, especially in competitive activity, where body balance is fundamental [11,12].

Competitive activity is activity which is practiced continuously and systematically, which involves intense psychophysical effort protracted over time. When the general tonic postural system and the craniomandibular system fail to adapt to excess functional demands, or can no longer withstand (e.g., due to overload or wear and tear) functional stresses, decompensation, and consequently pathology, occur [13]. Therefore, the psycho-physical conditions of athletes and their posture are important. Good posture results in reduced muscle work, less strain on ligaments, greater resistance to fatigue, and better management of movements [14,15,16]. Intense muscle exertions protracted over time, however, push the athlete to the limit, resulting in a framework of muscle fatigue that makes the balancing and coordination system less efficient, as well as a hydrosaline–hormonal imbalance, with a massive release of catecholamines and cortisol [3,17]. In addition, at the occlusal level, especially in endurance sports, or in those involving the use of maximum potential force, there is abnormal clenching of the dental arches (over-occlusion), resulting in excessive compression of the TMJ, which occurs as a physiological epiphenomenon during physical exertion. This non-physiological compression causes altered muscle contractions, and thus, increased fatigue, which is accentuated in cases of malocclusion [18].

In these cases, it is necessary to implement a therapy aimed at recovering and maintaining the athlete’s neuromuscular balance, which is also chosen through the use of instrumental methods, such as surface cranial electromyography, that allow a detailed study of these dysfunctions and a consequent therapeutic choice [2,19,20,21]. Electromyography (EMG) refers to a diagnostic and functional technique of recording and analyzing the myoelectric signal, or electrical biopotential, related to muscle activity during contraction. Depending on how the signal is recorded, it is generally referred to as surface electromyography (sEMG) or “needle” electromyography (also called electroneurography, ENG). sEMG consists of signal acquisition using electrodes placed on the skin (surface electrodes), while needle EMG involves the use of subcutaneous needles placed in direct contact with the muscle of interest (the needle will be longer the deeper the muscle). While needle electromyography allows analysis of a single motor unit, surface electrodes allow highlighting a group of motor units (with associated nerves) and their conduction velocity [12,22]. In a recent study, the use of electromyography during an orthodontic treatment allowed the register of changes in the activity of masseter and temporalis muscles that could influence the postural balance system [23,24]. The aim of this study was to use sEMG as an aid in the diagnosis and treatment of occlusal and muscular disorders and to value the effect on postural muscle balance.

## 2. Materials and Methods

### 2.1. Sample Description

A cross-sectional study was carried out on a unique sample of patients involved in professional sports. The study was conducted at the Faculty of Medical Sciences, University of Tetovo in North Macedonia. The sample consisted of 25 athletes, 20 male and 5 female, of different ages ranging from 16 to 37 years, all playing in Professional Sports Clubs engaged in the respective Italian National Championship. The study was conducted according to the guidelines of the Declaration of Helsinki and approved by the local Ethics Committee of the University of Tetovo, Tetovo, North Macedonia (Nr. 09-154/1 Prot.). Informed consent was obtained from the subjects involved in the study. Written informed consent has been obtained to publish this paper.

### 2.2. Study Protocol

Following the collection of anamnestic data and physical evaluation of the athletes, carried out by a physician and physiotherapist, they were divided into two groups:“Symptomatic patients” (13 patients) who reported, at the time of the examination, an injured condition or discomfort during the performance of competitive activity.“Asymptomatic patients” (12 patients) who did not present any muscle-osteo-articular discomfort in sports practice.

Subsequently, each athlete underwent a complete odontostomatologic examination and two additional subgroups could be identified:“Dysfunctional athletes” (6 patients) who were found to have craniomandibular disorders. The signs and symptoms considered to classify athletes as dysfunctional were as follows:-Pain in the preauricular area, TM joints, or chewing muscles;-Limitation or deviation of mandibular movement;-TM joint noises (clicking and crepitus) during mandibular function.“Functional athletes” (19 patients) who presented normal occlusion or malocclusion, but without any disorder related to it.

It is interesting to note that of the 25 athletes examined, most (20) were football players. Among the latter, 11 were symptomatic: 4 had pubalgia, 4 had leg muscular injuries, 2 had pelvic rotation, and 1 had back pain. However, among the football players, only 3 were classified as cranio-mandibular dysfunctional.

The characteristics of the sample study have been collected in Table 1.

This initial dental approach was followed by a global postural assessment in orthostasis, first examining the reciprocal position of the skeletal components and the muscle tone of the various body segments in stillness, paying more attention to those of the neck, head, and the relationship between them and the muscles of the scapular girdle, trunk, and pelvic girdle. This was followed by performing “anterior bending” tests (anterior flexion of the torso), with and without wax shims, which determined an initial estimate of the athlete’s overall degree of elasticity and the presence of any shortening of the muscle kinetic chains. Clinical assessment was followed by instrumental electromyography [25,26]. Cranial EMG (Teethan^®^) in this setting represents the quantitative aspect of the analysis and becomes the fundamental completion of the diagnostic process. The electromyographic recording procedure was carried out using two clamping tests: one in maximum intercuspation according to the athlete’s habitual occlusion and the other one by interposing 2 mm thick wax elevations between the arches (Figure 1).

Functional assessment index values were then determined, with the formulation of a percentage ferrule representing each athlete’s global index of cranial neuromuscular balance, thus allowing an overview of the patient’s musculo-occlusal condition with and without increased vertical dimension (Figure 2).

After analyzing these indices, it was possible to opt for the most valid therapeutic choice for each athlete to implement a treatment plan to solve the dysfunctional athlete’s symptoms. Therapy for cranio-mandibular disorders was the same for all patients and consists of the use of an occlusal splint. For each patient, the device was functionalized by discarding abnormal occlusal contacts in order to achieve balanced occlusion and neuromandibular balance. The use of electromyography allowed the splint to be functionalized based on muscle activity. Subsequently, one month after the start of the ortho-gnathological therapy, a follow-up was carried out to evaluate, by comparing the data with the initial data, the therapy’s effectiveness [27,28,29].

The steps for acquisition, visualization, and computer processing were reported, describing the procedures for recording in pre- and post-treatment ortho-gnathology, implementing everything in order to have a signal that is as interpretable as possible and with as few artifacts as possible.

### 2.3. Functional Assessment Indices

Analysis of electromyographic signals is traditionally qualitative, but the evolution in the knowledge of bioelectricity has allowed the development of quantitative methods for the interpretation of the electromyographic signal: functional indices. These indices derived from the ratio of average voltage measurements calculated over a given time interval (μV/μV and expressed in %), during the symmetrical contraction of homologous muscles.
POC (Percentage of Overlapping Coefficient):

POC is defined as an index of the symmetrical distribution of muscular activity determined by dental occlusion. It is related to the degree of overlapping of the activation curves of homologous muscles. The electromyographic potentials are expressed as a percentage of the maximum clamping voluntary (MVC) on wax, and the activation curves of the pairs of homologous muscles (Temporal “TA”–Masseter “MM”), are compared by calculating a muscle symmetry index capable of accounting for the entire morphology of the electromyographic signal. If each pair of homologous muscles contracted in perfect symmetry, the POC value would be equal to 100%. As is well known, however, by definition in nature, perfect symmetry does not exist, so values of POC are equal to or greater than the value of 83%.
BAR (Barycenter)

As the name suggests, this index assesses the position of the occlusal center of gravity. It is obtained by comparing the activities of temporalis and masseter muscles. Specifically, when the contact points tend to focus on the molars, the masseters register a greater contraction than the temporal corresponding (posterior barycenter). Conversely, in the occlusal condition in which the barycenter shifts to the anterolateral sectors (that is, to the first-second premolar), the temporals express greater contraction forces (anterior barycenter). In this case, there are a-physiological and bilateral temporomandibular joints overload, which can lead to pathological conditions. The BAR value, in an ideal “normal occlusion,” always expresses a prevalence of activity of the pair of masseter muscles over that of the temporalis, and thus an occlusal barycenter displacement posteriorly.
TORS (Mandibular Torsion Index)

This index assesses the torsion attitude of the mandible, in the horizontal plane, when it is occluded with the upper jaw. It is obtained by comparing the activities of the crossed muscles: in case of the prevalence of the right temporalis and left masseter pair, there is a rightward torsion; in case of the prevalence of the left temporal and right masseter, a torsion to the left is recorded. When the TORS index value is >90%, there is no mandibular torsion. In contrast, if this index is less than 90%, the muscles tend to twist the jaw, either to the right or to the left, depending on whether one or the other muscle pair prevails, based on the position of the occlusal fulcrums.

### 2.4. Statistical Analysis

Descriptive statistical analysis was performed on calculations of proportions, measures of central tendency, and variability for sociodemographic and clinical aspects. The data obtained were transformed into binary values (not effective = 0; effective = 1) for comparison. In addition, averages were evaluated for the values of each electromyographic index at t0 and t1. Chi-square test was used to establish the effectiveness of gnathological treatment in the sample examined, comparing the results obtained with those expected, given the impossibility of having a control group. In addition, t-test for paired data was performed to compare the averages obtained in the functional assessment indices at the first visit (t0) and in follow-up (t1) (*p* value ≤ 0.05).

## 3. Results

Splint treatment showed overall efficacy in 72% of cases versus 28% who did not respond to therapy (*p* = 0.028). In particular, it had statistically significant efficacy on rebalancing of the barycenter (BAR) in 76% (*p* = 0.028) of treated cases. However, the treatment did not appear to affect torsion (TORS) (*p* = 0.317).

Pre- and post-therapy measurements of electromyographic functional indices were reported in Table 2.

When comparing the mean values at t0 and t1, the t-test for paired data showed statistical significance in the difference of pre- (0.860 ± 0.124) and post-treatment (0.880 ± 0.114; *p* = 0.023) and torsion values in the pre- (0.822 ± 0.134) and post-bite (0.855 ± 0.150; *p* = 0.034) barycenter (Table 3) [30,31,32,33].

However, with a larger study sample, most likely, significance could have been obtained in the other values, as well, so the following paper could serve as a starting point for further study.

## 4. Discussion

Surface cranial electromyography allows for the assessment of occlusal status and quantification of neuromuscular balance; that is, understanding dental occlusion from a functional point of view. It represents, therefore, a diagnostic revolution because it makes it possible to see what until now was only perceptible by palpation, and therefore not quantifiable [34].

Several studies have established the reliability of sEMG when used therapeutically in the functional investigation of various muscle types. However, the authors claimed its low repeatability, which could be attributed to the presence of numerous variables, such as skin resistivity or the adipose layer thickness [34,35,36].

The objectives of the paper were achieved by analyzing the electromyograph and the electromyographic signal, first providing a general view of it and then applying it to the sports field. The present study corroborated the usefulness of in-sports practice, as it can quantify the extent and timing of muscle activation, at rest, and during physical activities, and the way they qualitatively affect movement through joint stabilization and coordination [37,38,39]. Consequences and perspectives of the data collected influence the possibility of therapeutic choices aimed at balancing the neuromotor component, such as the use of the bite, found to be effective in 72% of cases, and in the programming of work and recovery cycles tailored to the needs of each athlete, whether for rehabilitation purposes, for the purpose of implementing sports performance or, more simply, for the prevention of muscle damage.

In addition, based on the present study, the influence of dental occlusion on the muscle balance of the masticatory muscles and also on the activity of some postural muscles can be hypothesized.

A meta-analysis on the use of sEMG to evaluate relationships between masticatory muscles and postural muscles found that correlation between the masticatory system and muscle activity of other body districts can be experimentally detected using sEMG, but this correlation has poor clinical relevance [25].

However, Julià-Sánchez et al. found that dental occlusion affects the biomechanical and viscoelastic properties of the masticatory and postural muscles using the MyotonPRO^®^ system [6].

The influence of occlusal status on stability was also demonstrated in a paper by Heit et al. who found a significant increase in balance at rest rather than at maximum intercuspation [39].

Ferrillo et al., in their systematic review, sought to determine how occlusal splints affected the spinal posture of TMD patients. According to their study, the occlusal splint may be thought of as a non-invasive therapy strategy for TMD patients. Despite the limited evidence of proper inquiry for postural assessment, they discovered that occlusal splints may have a good influence on posture in TMD patients. To examine the effects of occlusal splints, additional research utilizing combined force platform stabilometry and kinematic measurement of the spine is needed [40].

These findings are consistent with prior study that used the sEMG in order to measure both the muscular balance of masticatory muscles and its influence on the activity of some postural muscles. It found a substantial reduction in postural muscle rest activity (sternocleidomastoid, erector spinae, and soleus) in participants with dental malocclusions after a neuromuscular occlusion balancing an acrylic wafer [41].

Finally, a systematic review and meta-analysis showed that rehabilitation approaches could be effective in reducing pain in patients with muscle-related TMJ disorders, despite the small number of RCTs [42].

However, Deregibus et al. in their work suggested that occlusal splints, independently from being built on the upper or lower arch, are not significantly effective in reducing pain in TMD patients [43].

The following limitations of the present study should be mentioned: the small sample number and one-month follow-up. This analysis may be a cue for further study on the subject, confident that with a larger sampling, the data would present a greater overall significance. Furthermore, to evaluate the effectiveness of EMG, a future study should rather plan to compare the sEMG with another tool.

From this study, the sEMG method resulted useful in the most appropriate therapeutic choice for each athlete, but above all, its usefulness is expressed in the monitoring of the treatment, and in the possibility of modifying it in the course of the treatment when the results obtained are not in line with our expectations. This study did not intend to demonstrate the absolute efficacy of the electromyographic technique, but we highlight how this method can provide us with a reliable additional indication regarding the choice of the most appropriate therapeutic course. This is particularly true in the case of professional athletes susceptible to imbalances and imbalances due to assiduity in sports practice, in which re-establishing congruous occlusal contact is equivalent to improving the degree of cranial stabilization and, consequently, to provoking a descending cascade of neuromuscular balance.

## 5. Conclusions

The use of cranial sEMG in competitive athletes, therefore, is clinically indicated in the therapeutic choice aimed at balancing occlusal loads and improving the overall tonic postural attitude of the individual, resulting in positive feedback in the qualitative assessment of sports performance.

## Figures and Tables

**Figure 1 ijerph-19-07975-f001:**
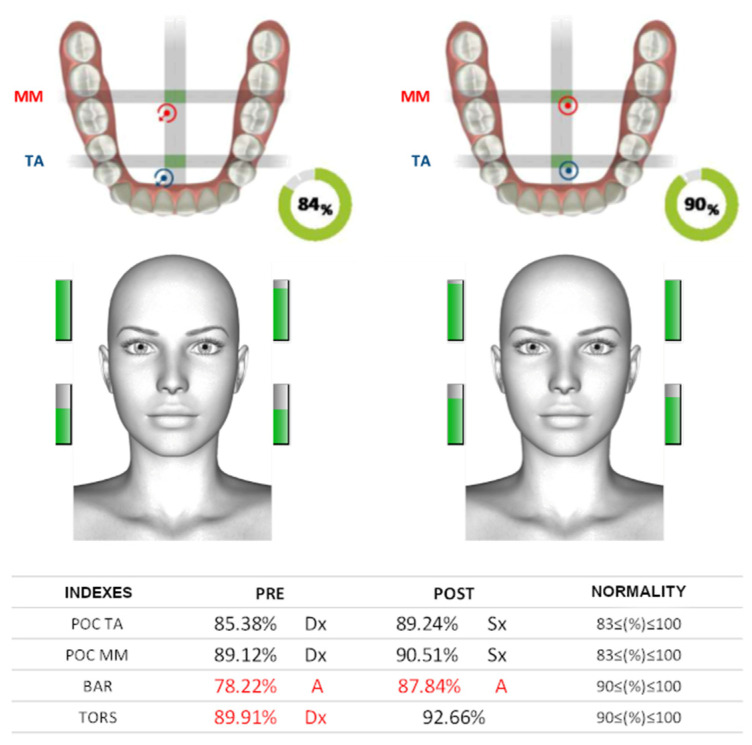
Example of recording electromyographic indices. In the graphic representation of the occlusal board, the position of the targets summarizes, visually, all the calculated occlusal indices and the intensity of the contraction of each muscle by green bars. Values that were outside the normal range are shown in red.

**Figure 2 ijerph-19-07975-f002:**
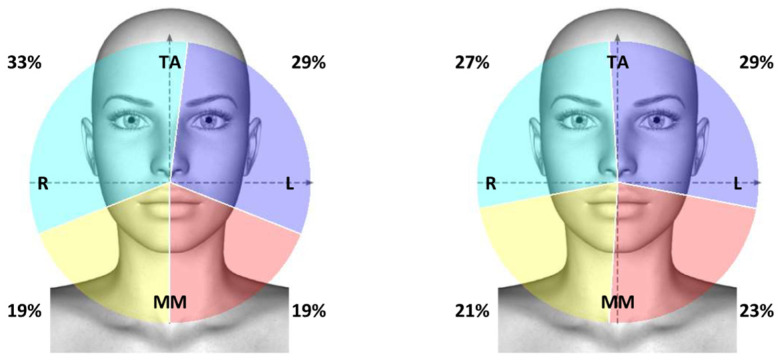
Graphic representation of the analyzed muscles, indicated by light blue and purple colors referring to the activity of right (R) and left (L) anterior temporalis muscles (TA), respectively, and by yellow and red referring to the activity of right (R) and left (L) masseter muscles (MM), respectively.

**Table 1 ijerph-19-07975-t001:** Study sample.

#	Age	Sex	Sport Practiced	Symptomatic Athletes	Asymptomatic Athletes	Dysfunctional Athletes	Functional Athletes
1	25	M	Judo		Sport performance		Malocclusion
2	27	F	Football				Normocclusion
3	18	F	Dance		Sport performance	Malocclusion, TMJ, and auricular area pain	
4	27	M	Football	Hip flexor muscle inflammation		Malocclusion, TMJ click, and chewing muscles contraction	
5	19	M	Judo		Sport performance	Malocclusion, TMJ, and facial massif pain	
6	15	M	Football	Pubalgia			Normocclusion
7	15	M	Football	Pubalgia			Normocclusion
8	31	M	Football		Sport performance		Malocclusion
9	26	M	Football	Lumbar muscles contraction			Normocclusion
10	18	F	Ping pong	Epicondylitis			Normocclusion
11	16	M	Football		Sport performance		Malocclusion
12	19	M	Football		Sport performance		Malocclusion
13	29	M	Football	Pubalgia			Normocclusion
14	21	M	Football	Pelvic rotation	Sport performance		Normocclusion
15	34	M	Football		Sport performance		Normocclusion
16	16	M	Football	Pelvic rotation	Sport performance		Normocclusion
17	27	M	Football	Quadriceps muscles strain			Normocclusion
18	33	M	Football		Sport performance	Mandibular fracture	
19	21	F	Dance		Sport performance	TMJ pain	
20	19	M	Football	Abductor muscles strain			Normocclusion
21	24	M	Football	Pubalgia			Normocclusion
22	16	M	Football		Sport performance		Normocclusion
23	37	M	Football	Quadriceps muscles strain			Malocclusion
24	25	M	Football		Sport performance	TMJ click and chewing muscles contraction	
25	32	F	Volleyball	Rotator cuff muscles strain			Malocclusion

**Table 2 ijerph-19-07975-t002:** PRE and POST treatment indices.

#	POC TA PRE	POC TA POST	Improvement	BAR PRE	BAR POST	Improvement	TORS PRE	TORS POST	Improvement
1	0.811	0.818	1	0.750	0.905	1	0.881	0.892	1
2	0.853	0.892	1	0.782	0.878	1	0.899	0.927	1
3	0.872	0.859	0	0.821	0.852	1	0.921	0.889	1
4	0.991	0.898	0	0.883	0.969	1	0.934	0.916	0
5	0.837	0.759	1	0.854	0.872	1	0.884	0.849	0
6	0.871	0.851	0	0.915	0.916	1	0.919	0.913	1
7	0.878	0.884	1	0.898	0.916	1	0.900	0.919	0
8	0.774	0.856	1	0.826	0.866	1	0.772	0.885	1
9	0.836	0.880	1	0.257	0.165	0	0.297	0.341	0
10	0.482	0.868	1	0.841	0.746	1	0.818	0.920	1
11	0.903	0.895	0	0.691	0.931	1	0.906	0.908	0
12	0.880	0.904	1	0.854	0.899	1	0.925	0.922	0
13	0.887	0.883	0	0.873	0.931	1	0.813	0.928	1
14	0.865	0.887	1	0.788	0.827	1	0.836	0.862	1
15	0.838	0.842	1	0.865	0.882	1	0.891	0.903	1
16	0.882	0.867	0	0.886	0.851	0	0.916	0.909	0
17	0.866	0.867	1	0.914	0.896	1	0.918	0.889	1
18	0.876	0.874	0	0.913	0.913	0	0.899	0.904	0
19	0.840	0.830	0	0.849	0.873	0	0.886	0.913	1
20	0.890	0.887	0	0.862	0.875	1	0.909	0.913	0
21	0.866	0.862	0	0.899	0.926	1	0.875	0.911	1
22	0.893	0.898	1	0.913	0.909	0	0.913	0.909	0
23	0.845	0.846	1	0.687	0.848	1	0.846	0.909	1
24	0.776	0.812	1	0.896	0.901	1	0.864	0.870	1
25	0.644	0.701	1	0.834	0.826	0	0.885	0.903	1

**Table 3 ijerph-19-07975-t003:** Statistical analysis of the differences of mean values at t0 and t1. Statistically significant values are highlighted in bold type. Statistical significance is evident in the difference in the values of: -TOR PRE (0.860 ± 0.124) and POST (0.880 ± 0.114; *p* = 0.023); -BAR PRE (0.822 ± 0.134) and POST (0.855 ± 0.150; *p* = 0.034).

		Paired Differences					t	df	Sig. (2-Tailed)
		Mean	Std. Deviation	Std. Error Mean	95% Confidence Interval of the Difference				
					Lower	Upper			
Pair 1	POC TA pre bite—POC TA post bite	−0.01856	0.084665	0.016933	−0.053508	0.016388	−1.096	24	0.284
Pair 2	BAR pre bite—BAR post bite	**−0.03288**	0.073074	0.014615	−0.063044	−0.002716	−2.25	24	**0.034**
Pair 3	TORSIO N PRE BITE—TORSIO N POST BITE	**−0.01988**	0.04106	0.008212	−0.036829	−0.002931	−2.421	24	**0.023**

## Data Availability

All experimental data to support the findings of this study are available contacting the corresponding author upon request.

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
