# Peer review of "Experimental Analysis of the Use of Cranial Electromyography in Athletes and Clinical Implications"

_ijerph, 2022, doi:10.3390/ijerph19137975_

Round 1
Reviewer 1 Report
Thank you for the opportunity to evaluate the manuscript: "Experimental analysis of the use of cranial electromyography in athletes and clinical implications."
- In the Abstract you use the terms "afunctional" and "dysfunctional athletes". If they are the same, I am asking for unification. If different, please explain what you mean by dysfunctional athletes.
- In the Abstract you write about "orthognathic therapy", which for me clearly indicates the treatment of skeletal malocclusion with osteotomy and fixation of bone fragments. Later you write about the use of occlusal splints. For the sake of clarity, I suggest that you write in the abstract that splint therapy was used.
- In the Abstract you use the term "functional athletes", in Methods "healthy athletes". If they are the same, I would like to ask for unification, and if they are different, then for full explanations of these terms in the Methods.
- The number of "healthy athletes" in the text is different from the number of "functional athletes" in Table 1. In the course of corrections, pay attention to the size of this group described in the Abstract.
- You write that you have made the qualifications to the groups "dysfunctional athletes" and "healthy athletes" on the basis of a thorough study. Describe the details of this study. If possible, indicate the numeric variables and attach them to the results.
- On the basis of what criteria did you qualify the subjects as "dysfunctional athletes" and "healthy athletes"? Please indicate the appropriate classification of dysfunctions or the previously published qualification protocol or add to the content of this article the details of the qualification protocol, if it was original.
- Your research group is heterogeneous in terms of the sports practiced, with one dominant. In the course of the evaluation of the results, please do an analysis for the football subgroup.
- Please describe in more detail what treatment the study group was subjected to. If it was different for each person, add the appropriate column in table 1. If it was the same, please write what splints were used and whether it was the only intervention. Determine if there were other forms of treatment in addition, e.g. pharmacotherapy, intramuscular or intra-articular injections.
- Not all abbreviations appearing in the figures are explained at the end of the manuscript. For ease of reading by the reader, consider further elaborating on the abbreviations of a given figure in its description, or simply do not use abbreviations (especially in the Indexes column in Fig. 1).
- Consider making the gray background in Figure 1 transparent or white.
- In the description of Figure 2 you explain the use of red and blue. You are probably assigning them inversely to muscles by mistake - please verify that. Also add a description of how to interpret yellow and purple.
- In the discussion, make sure about the spelling, because I saw the spelling "MyotonProÒ", where the penultimate letter is not capitalized.
- In the Aim section (in your manuscript as part of the introduction, which is also correct), ask the correct research question and answer it in the Conclusions. Currently, in the Aim section, you write about the use of sEMG, and you should rather plan to confront the sEMG with another tool or plan to show differences in sEMG recordings before and after therapy (which you did). The conclusions are too broad and mostly consist of content that can be successfully transferred to the Discussion section. Let your conclusions be based on a simple answer to a research question and a summary of the analysis of your results in the context of the research of other authors (conclusions from the Discussion).
- In the References section, the names of the authors of this article are repeated many times. Many of the items in my opinion are not sufficiently related to the subject of this manuscript. Presumably, items 2 and 27 are the same. It is possible that there are more issues in this section that I did not catch. I urge that this section be completely re-edited.
In conclusion, I think the research idea was good, but the study group was too small and too heterogeneous. There are methodological errors in the manuscript that must be corrected before publication. There are numerous minor errors in the content, some of which I have indicated. The large number of self-citations is very disturbing to me and I am concerned about the scientific honesty of the Authors.
Author Response
REVIEWER 1
Thank you for the opportunity to evaluate the manuscript: "Experimental analysis of the use of cranial electromyography in athletes and clinical implications."
- In the Abstract you use the terms "afunctional" and "dysfunctional athletes". If they are the same, I am asking for unification. If different, please explain what you mean by dysfunctional athletes.
They are the same. The term “afunctional” has been correct to “dysfunctional”.
- In the Abstract you write about "orthognathic therapy", which for me clearly indicates the treatment of skeletal malocclusion with osteotomy and fixation of bone fragments. Later you write about the use of occlusal splints. For the sake of clarity, I suggest that you write in the abstract that splint therapy was used.
It has been corrected.
- In the Abstract you use the term "functional athletes", in Methods "healthy athletes". If they are the same, I would like to ask for unification, and if they are different, then for full explanations of these terms in the Methods.
They are the same. The term “healthy” has been correct to “functional”.
- The number of "healthy athletes" in the text is different from the number of "functional athletes" in Table 1. In the course of corrections, pay attention to the size of this group described in the Abstract.
The term “healthy” has been correct to “functional”. The number has been corrected in the text, in the abstract and in the table.
- You write that you have made the qualifications to the groups "dysfunctional athletes" and "healthy athletes" on the basis of a thorough study. Describe the details of this study. If possible, indicate the numeric variables and attach them to the results.
Athletes with the following signs and symptoms at clinical examination were considered "dysfunctional":
- Pain in the preauricular area, TM joints, or chewing muscles;
- Limitation or deviation of mandibular movement;
- TM joint noises (clicking and crepitus), during mandibular function.
- On the basis of what criteria did you qualify the subjects as "dysfunctional athletes" and "healthy athletes"? Please indicate the appropriate classification of dysfunctions or the previously published qualification protocol or add to the content of this article the details of the qualification protocol, if it was original.
Athletes with the following signs and symptoms were considered "dysfunctional":
- Pain in the preauricular area, TM joints, or chewing muscles;
- Limitation or deviation of mandibular movement;
- TM joint noises (clicking and crepitus), during mandibular function.
- Your research group is heterogeneous in terms of the sports practiced, with one dominant. In the course of the evaluation of the results, please do an analysis for the football subgroup.
As you pointed out, of the 25 athletes examined, most (20) were football players. Among the latter, 11 were symptomatic. Specifically, 4 had pubalgia, 4 leg muscular injuries, 2 pelvic rotation and 1 back pain. Among the football players, only 3 were classified as cranio-mandibular dysfunctional.
These data have been added to lines 135-138.
We also incorporated the other athletes in our study because the sample of football players would have been small for an analysis. In addition, among the other 5 athletes, 3 were dysfunctional. Thanks to your suggestion, we can carry out a further study by focusing only on the football players with a larger sample.
- Please describe in more detail what treatment the study group was subjected to. If it was different for each person, add the appropriate column in table 1. If it was the same, please write what splints were used and whether it was the only intervention. Determine if there were other forms of treatment in addition, e.g. pharmacotherapy, intramuscular or intra-articular injections.
Therapy for cranio-mandibular disorders has been the same for all patients and consists of the use of an occlusal splint. For each patient, the device was functionalized by discarding at abnormal occlusal contacts in order to achieve balanced occlusion and neuromandibular balance.
- Not all abbreviations appearing in the figures are explained at the end of the manuscript. For ease of reading by the reader, consider further elaborating on the abbreviations of a given figure in its description, or simply do not use abbreviations (especially in the Indexes column in Fig. 1).
The figures have been modified. The abbreviations have been added.
- Consider making the gray background in Figure 1 transparent or white.
Done
- In the description of Figure 2 you explain the use of red and blue. You are probably assigning them inversely to muscles by mistake - please verify that. Also add a description of how to interpret yellow and purple.
It has been corrected.
- In the discussion, make sure about the spelling, because I saw the spelling "MyotonProÒ", where the penultimate letter is not capitalized.
It has been corrected.
- In the Aim section (in your manuscript as part of the introduction, which is also correct), ask the correct research question and answer it in the Conclusions. Currently, in the Aim section, you write about the use of sEMG, and you should rather plan to confront the sEMG with another tool or plan to show differences in sEMG recordings before and after therapy (which you did). The conclusions are too broad and mostly consist of content that can be successfully transferred to the Discussion section. Let your conclusions be based on a simple answer to a research question and a summary of the analysis of your results in the context of the research of other authors (conclusions from the Discussion).
Thank you for your suggestions. The conclusions have been modified.
- In the References section, the names of the authors of this article are repeated many times. Many of the items in my opinion are not sufficiently related to the subject of this manuscript. Presumably, items 2 and 27 are the same. It is possible that there are more issues in this section that I did not catch. I urge that this section be completely re-edited.
References have been modified. Many references have been removed.
In conclusion, I think the research idea was good, but the study group was too small and too heterogeneous. There are methodological errors in the manuscript that must be corrected before publication. There are numerous minor errors in the content, some of which I have indicated. The large number of self-citations is very disturbing to me and I am concerned about the scientific honesty of the Authors.
Thank you for your suggestions. The article has been improved.
Reviewer 2 Report
Dear authors,
I consider that the article entitled “Experimental analysis of the use of cranial electromyography in athletes and clinical implications” addresses an extremely complex topic that requires an interdisciplinary approach.
In this paper was evaluated the role of surface cranial electromyography in the diagnosis and treatment of occlusal and muscular disorders in professional athletes.
The authors wrote in:
2. Materials and Methods
2.1 Sample description
A cross-sectional study (cross-sectional study) was carried out on a unique sample of patients involved in professional sports. The study was conducted at the Faculty of Medical Sciences, University of Tetovo in North Macedonia. The sample consisted of 25 athletes, 20 male, and 5 female, having different ages ranging from 16 to 37 years, all playing in Professional Sports Clubs engaged in the respective Italian National Cham- pionship.
Here, the authors must specify:
The study was conducted according to the guidelines of the Declaration of Helsinki and approved by the local Ethics Committee of University of Tetovo, Tetovo, North Macedonia (Nr. 09-154/1 Prot.).
Informed consent was obtained from the subjects involved in the study. Written informed consent has been obtained from the patient to publish this paper.
Author Response
REVIEWER 2
Dear authors,
I consider that the article entitled “Experimental analysis of the use of cranial electromyography in athletes and clinical implications” addresses an extremely complex topic that requires an interdisciplinary approach.
In this paper was evaluated the role of surface cranial electromyography in the diagnosis and treatment of occlusal and muscular disorders in professional athletes.
The authors wrote in:
- Materials and Methods
2.1 Sample description
A cross-sectional study (cross-sectional study) was carried out on a unique sample of patients involved in professional sports. The study was conducted at the Faculty of Medical Sciences, University of Tetovo in North Macedonia. The sample consisted of 25 athletes, 20 male, and 5 female, having different ages ranging from 16 to 37 years, all playing in Professional Sports Clubs engaged in the respective Italian National Cham- pionship.
Here, the authors must specify:
The study was conducted according to the guidelines of the Declaration of Helsinki and approved by the local Ethics Committee of University of Tetovo, Tetovo, North Macedonia (Nr. 09-154/1 Prot.).
Informed consent was obtained from the subjects involved in the study. Written informed consent has been obtained from the patient to publish this paper.
It has been added also in materials and methods.
Reviewer 3 Report
Dear Authors,
I have read your paper with great interest and attention. Your paper is focused on the use of cranial electromyography in athletes and its possible implication on occlusal and muscular disorders in professional athletes. The topic is very current and its implication in athletes performance can play a crucial role in the management of occlusion disorders in athletes. After a careful reading, I can affirm that the paper is cleary written and succinct. Nevertheless, I have some critical issue to address:
Introduction: The authors affirm that in literature there are discordant data regarding the correlation between occlusal disorder and posture. That’s true, but in the last years different studies have investigate this association and possible correlation between Spinal posture and dental occlusion. Please, read “Ferrillo M, Marotta N, Giudice A, Calafiore D, Curci C, Fortunato L, Ammendolia A, de Sire A. Effects of Occlusal Splints on Spinal Posture in Patients with Temporomandibular Disorders: A Systematic Review. Healthcare (Basel). 2022 Apr 15;10(4):739. doi: 10.3390/healthcare10040739”. The authors aiming to define the EMG as tool in the diagnosis and treatment of occlusal and muscle disorders, but how do you want define it? Briefly, I think that the study design is not appropriate, because there is not a specific control group or another time of the study of the same cohort. Please, specify the choice of that and modify the aim of the study.
Methods: There are different points to stress:
1) Who have diagnosed the muscle/joint injuries in the athletes?
2) Please, specify the exact diagnosis of osteoarticular disease. It’s not clear the localization and the severity of the injury. At the same time, specify the type of the occlusal disorders. I suggest to sort this information in another table or rewrite the table 1
3) The authors talks about an orthognatology treatment, but this part is not clear. Which kind of treatment athletes underwent?
4) Please, specify the name and the production company of the EMG
Results: Please, rewrite the table 3, adding the statistical significativities in table with marks
Discussion: The discussion should be rewrite focusing the attention on the possible therapeutic option for spine control in different field (rehabilitation, orthognatology). Please, read “Deregibus A, Ferrillo M, Grazia Piancino M, Chiara Domini M, de Sire A, Castroflorio T. Are occlusal splints effective in reducing myofascial pain in patients with muscle-related temporomandibular disorders? A randomized-controlled trial. Turk J Phys Med Rehabil. 2021 Mar 4;67(1):32-40. doi:10.5606/tftrd.2021.6615” and “Ferrillo M, Ammendolia A, Paduano S, Calafiore D, Marotta N, Migliario M, Fortunato L, Giudice A, Michelotti A, de Sire A. Efficacy of rehabilitation on reducing pain in muscle-related temporomandibular disorders: A systematic review and meta-analysis of randomized controlled trials. J Back Musculoskelet Rehabil. 2022 Feb 18. doi: 10.3233/BMR-210236.” Moreover, the study of Bergamini et al. (references 53) is not clear explained. What’s mean that they found a substantial reduction in postural muscle rest activity in participants with dental malocclusions after a neuromuscular occlusion balancing to reach an optimal occlusal relationship mandibular coronoid process hypertrophy? Please specify
Best Regards
Author Response
REVIEWER 3
Dear Authors,
I have read your paper with great interest and attention. Your paper is focused on the use of cranial electromyography in athletes and its possible implication on occlusal and muscular disorders in professional athletes. The topic is very current and its implication in athletes performance can play a crucial role in the management of occlusion disorders in athletes. After a careful reading, I can affirm that the paper is cleary written and succinct. Nevertheless, I have some critical issue to address:
Introduction: The authors affirm that in literature there are discordant data regarding the correlation between occlusal disorder and posture. That’s true, but in the last years different studies have investigate this association and possible correlation between Spinal posture and dental occlusion. Please, read “Ferrillo M, Marotta N, Giudice A, Calafiore D, Curci C, Fortunato L, Ammendolia A, de Sire A. Effects of Occlusal Splints on Spinal Posture in Patients with Temporomandibular Disorders: A Systematic Review. Healthcare (Basel). 2022 Apr 15;10(4):739. doi: 10.3390/healthcare10040739”. The authors aiming to define the EMG as tool in the diagnosis and treatment of occlusal and muscle disorders, but how do you want define it? Briefly, I think that the study design is not appropriate, because there is not a specific control group or another time of the study of the same cohort. Please, specify the choice of that and modify the aim of the study.
The paper you suggested has been added in the discussion section (lines 375-381).
Aim has been modified.
We also added in the Discussion section:
This study did not intend to demonstrate the absolute efficacy of the electromyographic technique, but we highlight how this method can provide us with a reliable additional indication regarding the choice of the most appropriate therapeutic course.
This analysis may be a cue for further studies on the subject a larger sample and a control group. Furthermore, to evaluate the effectiveness of EMG may be of interest you should rather plan to compare the sEMG with another tool.
Methods: There are different points to stress:
- Who have diagnosed the muscle/joint injuries in the athletes?
Physical examination of the athletes was carried out by a physician and physiotherapist. (LINE 116)
Please, specify the exact diagnosis of osteoarticular disease. It’s not clear the localization and the severity of the injury. At the same time, specify the type of the occlusal disorders. I suggest to sort this information in another table or rewrite the table 1
Table has been modified and integrated
- The authors talks about an orthognatology treatment, but this part is not clear. Which kind of treatment athletes underwent?
Therapy for cranio-mandibular disorders has been the same for all patients and consists of the use of an occlusal splint. For each patient, the device was functionalized by discarding at abnormal occlusal contacts in order to achieve balanced occlusion and neuromandibular balance. - Please, specify the name and the production company of the EMG.
The production company of the EMG was TeethanÒ - Results: Please, rewrite the table 3, adding the statistical significativities in table with marks.
Table has been modified - Discussion: The discussion should be rewrite focusing the attention on the possible therapeutic option for spine control in different field (rehabilitation, orthognatology). Please, read “Deregibus A, Ferrillo M, Grazia Piancino M, Chiara Domini M, de Sire A, Castroflorio T. Are occlusal splints effective in reducing myofascial pain in patients with muscle-related temporomandibular disorders? A randomized-controlled trial. Turk J Phys Med Rehabil. 2021 Mar 4;67(1):32-40. doi:10.5606/tftrd.2021.6615” and “Ferrillo M, Ammendolia A, Paduano S, Calafiore D, Marotta N, Migliario M, Fortunato L, Giudice A, Michelotti A, de Sire A. Efficacy of rehabilitation on reducing pain in muscle-related temporomandibular disorders: A systematic review and meta-analysis of randomized controlled trials. J Back Musculoskelet Rehabil. 2022 Feb 18. doi: 10.3233/BMR-210236.”
Discussion section has been improved and this papers have been added (lines 388-392).
Moreover, the study of Bergamini et al. (references 53) is not clear explained. What’s mean that they found a substantial reduction in postural muscle rest activity in participants with dental malocclusions after a neuromuscular occlusion balancing to reach an optimal occlusal relationship mandibular coronoid process hypertrophy? Please specify
The concept has been better explained (lines 382-387).
Round 2
Reviewer 1 Report
Thank you for responding to my comments.
I consider the corrections made to be sufficient.
Reviewer 3 Report
Dear Authors,
thank you for your revision. At the light of the new version, the paper is suitable for fully publication
Best regards